# Potential Therapeutic Effect of Continuous Positive Airway Pressure on Laryngopharyngeal Reflux in Obstructive Sleep Apnea Patients

**DOI:** 10.3390/jcm10132861

**Published:** 2021-06-28

**Authors:** Jae Hyuk Choi, Eunkyu Lee, Sang Duk Hong, Seung Kyu Chung, Yong Gi Jung, Hyo Yeol Kim

**Affiliations:** Samsung Medical Center, Department of Otorhinolaryngology-Head and Neck Surgery, School of Medicine, Sungkyunkwan University, Seoul 06351, Korea; chlwogurr@naver.com (J.H.C.); eunkyu2.lee@samsung.com (E.L.); sangduk.hong@samsung.com (S.D.H.); eskei.chung@samsung.com (S.K.C.)

**Keywords:** obstructive sleep apnea, continuous positive airway pressure, laryngopharyngeal reflux

## Abstract

To investigate the potential therapeutic effect of continuous positive airway pressure (CPAP) treatment on laryngopharyngeal reflux in obstructive sleep apnea (OSA) patients, we performed a retrospective analysis of data prospectively collected from patients who underwent CPAP therapy after being diagnosed with moderate to severe OSA between January 2019 and May 2020. Subjects were asked to complete the reflux symptom index (RSI) questionnaire before and after CPAP. Additionally, a laryngoscopic examination was performed to evaluate objective endoscopic findings and determine reflux finding score (RFS). A total of 46 patients were included in the analysis. Overall, significant decreases in mean RSI score (10.85 ± 6.40 vs. 8.80 ± 7.99, *p* < 0.001) and RFS (7.41 ± 3.32 vs. 4.65 ± 2.12, *p* < 0.001) were observed after CPAP treatment. Within subdomains of the RSI, throat clearing, postnasal drip, breathing difficulty, troublesome cough, and foreign body sensation were significantly improved by CPAP treatment. All subdomains of RFS, with the exception of posterior commissure hypertrophy and granuloma, showed significant differences after CPAP treatment. There were no differences between subgroups according to body mass index or severity of OSA. CPAP treatment in OSA potentially reduces laryngeal reflux symptoms and improves laryngeal examination findings.

## 1. Introduction

Obstructive sleep apnea (OSA) is a sleep-related disorder associated with upper airway collapse and reduction of airway flow [1]. OSA increases the risk of hypertension, cardiovascular events, cerebrovascular accidents, cognitive deterioration, and motor vehicle accidents [2]. In gastroenterology, OSA is related to liver injury, fatty liver disease, peptic ulcer, and reflux diseases such as gastroesophageal reflux and laryngopharyngeal reflux (LPR) [3].

LPR is a disease in which leakage of gastric acid from the upper esophageal sphincter damages the laryngopharyngeal mucosa. The main symptoms of LPR include hoarseness, throat clearing, dysphagia, and globus sensation. On endoscopy in patients with LPR, diffuse laryngeal edema, erythema, posterior commissure hypertrophy, and granuloma are generally observed [4]. Diagnosis of LPR is usually based on reflux symptom index (RSI) and reflux finding score (RFS). An RSI score >13 or RFS >7 is indicative of LPR [5,6].

A high incidence of LPR in OSA patients was revealed in several studies. The incidence of LPR is approximately 10% in the general population, whereas in OSA patients, the incidence ranged from 30.6% to 89.2% [7]. A few studies have suggested that OSA treatment relieves LPR. RSI improved in OSA patients after multilevel OSA surgery in one study [8]. Chronic cough and 24 h acid contact time improved with CPAP treatment [9,10]. However, to date, no studies have investigated the effects of CPAP treatment for OSA on LPR. The aim of this study was to evaluate change in laryngeal reflux symptoms after CPAP for the treatment of OSA, based on patient responses to the RSI questionnaire and RFS.

## 2. Materials and Methods

### 2.1. Study Subjects

Medical records of patients who used CPAP after diagnosis with moderate to severe OSA at the Department of Otorhinolaryngology-Head and Neck Surgery, Samsung Medical Center, between January 2019 and May 2020, were retrospectively reviewed. Due to the small number of patients who underwent CPAP treatment, mild OSA patients were excluded from this study. Overall, 476 moderate to severe OSA patients were enrolled. Patients who took medication for LPR, who underwent surgery before 3 months of CPAP treatment, who did not complete RSI questionnaires, or who did not undergo endoscopic exam of the larynx, or who had insufficient endoscopic results were excluded. The age, weight, height, body mass index (BMI), polysomnography, apnea–hypopnea index (AHI) after CPAP, and average CPAP usage time were also reviewed.

### 2.2. RSI Questionnaire and RFS

The RSI questionnaire, which contains nine questions, and endoscopic evaluation of RFS, which contains eight items, were undertaken in moderate to severe OSA patients who underwent CPAP therapy. The RSI is a questionnaire with nine questions created to document symptoms and the severity of LPR. Patients fill out the questionnaire on a scale of zero to five describing how these nine problems affected them over the past month. The maximum total score is 45, and a total score greater than 13 is considered indicative of LPR. As shown in Table 1, the RSI questionnaire was translated into Korean and used to evaluate patient symptom history and symptom characteristics [5]. The RFS is characterized using an eight-item evaluation sheet designed to assess the clinical severity of LPR based on laryngoscope findings. RFS rates the location and severity of inflammatory changes, including subglottic edema, ventricular obliteration, erythema, vocal cord edema, diffuse laryngeal edema, posterior commissure hypertrophy, granuloma, and thick endolaryngeal mucus, as shown in Table 2 [6]. The maximum total score is 26 points, and generally, a score of 7 points or more is considered indicative of LPR. To assess the potential adjuvant therapeutic effect of CPAP treatment on LPR, we compared RSI and RFS pretreatment and three months post-treatment.

### 2.3. Data Analysis

Statistical analysis was carried out using SPSS software version 27. Paired T-test was used to compare RFS and RSI scores, as well as subdomains of RSI and RFS. For further analysis, patients were divided into a moderate OSA group and a severe OSA group. Additionally, based on BMI, we divided patients into three groups: the normal group (BMI < 25), the overweight group (25 ≤ BMI < 30), and the obese group (BMI ≥ 30). Kruskal–Wallis test was used to compare these groups.

As reflux finding score is subjective, there may be differences in scoring between observers. To ensure the data were as objective as possible, three otolaryngologists evaluated the endoscopic images without knowing whether they were taken before or after treatment. The kappa value was calculated to determine inter-rater reliability.

## 3. Results

A total of 46 patients were included in the analysis. The mean age of patients was 52.7 ± 11.6 years old. Of those included, 43 patients were male and 3 were female. Pretreatment mean AHI score was 49.5 ± 22.7/h, and mean BMI was 28.6 ± 7.2 kg/m^2^. Among them, 12 people exceeded 13 points on the RSI score, and 26 exceeded 7 points on RFS (Table 3).

After CPAP treatment, mean AHI was decreased to 2.27 ± 1.80/h. The average CPAP usage time of patients was 337 min, and 42 patients (88.9%) had good compliance with an average usage time of 4 h or more. RSI score was significantly decreased after CPAP use (10.85 ± 6.40 vs. 8.80 ± 7.99, *p* < 0.001). RFS, which was initially 7.41 ± 3.32, improved after CPAP treatment to 4.65 ± 2.12 (*p* < 0.001). The number of patients exceeding the RSI score cutoff of 13 points decreased from 12 to 5 after treatment, and the number of patients exceeding the RFS cutoff of 7 points decreased from 26 to 9.

Within each subdomain of the RSI questionnaire, throat clearing, postnasal drip, breathing difficulty, troublesome cough, and foreign body sensation showed significant differences after CPAP treatment. Most subdomains of RFS, with the exception of posterior commissure hypertrophy and granuloma, showed significant differences after CPAP treatment (Table 4 and Table 5).

Differences in RSI and RFS between moderate OSA and severe OSA were also analyzed. In total, 12 patients had moderate OSA, and 34 had severe OSA. RSI score differences were 4.25 ± 4.94, 3.50 ± 3.54, and RFS differences were 3.33 ± 2.57 and 2.70 ± 2.48 in each group. There was no significant difference between the two groups within RSI score differences (*p* = 0.37) or RFS differences (*p* = 0.99). In addition, patients were divided into normal, overweight, and obese groups by BMI, but there were no significant differences in RSI score differences or RFS differences between the three groups (Table 6).

## 4. Discussion

In this study, we compared symptoms of LPR in patients with moderate to severe OSA using an RSI questionnaire and endoscopic findings before and after CPAP treatment. This study found that the total RSI score and RFS improved significantly after CPAP treatment in OSA patients. Some individual variables also significantly improved after CPAP treatment. To the best of our knowledge, there have been no previous studies investigating the effectiveness of CPAP for laryngopharyngeal reflux through both questionnaires and endoscopic findings.

The high incidence of LPR in patients with OSA has been confirmed in several studies, but the mechanism has not yet been established. In studies on gastroesophageal reflux disease (GERD) in OSA patients, some studies have explored the mechanism with 24 pH monitoring, transient lower esophageal sphincter relaxation, and intra-esophageal pressure, but no clear association was found [11]. Additionally, in LPR, unlike GERD, the upper esophageal sphincter (UES) is also involved. There is a hypothesis explaining the association between LPR and OSA, in which increased respiratory effort generates more negative intrathoracic pressure, increasing the transdiaphragmatic pressure gradient and contributing to reflux of gastric contents. However, studies on this suggest that UES pressure is increased in OSA patients, compared to the normal group, and there was no difference in LPR events compared with the normal group [12]. Thus, the mechanism for this relationship has not been revealed.

Typical treatments for LPR include diet control, weight loss, smoking and alcohol cessation, behavioral treatments such as not lying down immediately after eating, and proton pump inhibitor (PPI) treatment [13]. However, the effectiveness of PPIs in LPR treatment has been controversial. One study showed no statistically significant differences in the severity or frequency of reflux symptoms between patients receiving PPIs and placebo treatment among six randomized controlled trials [14].

Several studies explored the relationship between OSA and reflux disease and showed that reflux disease improved after CPAP therapy [15]. Our study also showed that the RSI score and RFS improved significantly after CPAP treatment. Throat clearing, postnasal drip, breathing difficulty, troublesome cough, and foreign body sensation were significantly improved in the RSI questionnaire after CPAP use. All subdomains of RFS showed significant improvement except for posterior commissure hypertrophy and granuloma. Our research team hypothesized that posterior commissure hypertrophy and granuloma are chronic changes and might not show improvement within a short treatment period of 3 months.

These results might result from various effects of CPAP on airway mucosa. First, using CPAP can reduce patient arousal and movement during sleep. Reducing arousal and movement can prevent changes in abdominal pressure and thus reduce reflux events [16]. However, as mentioned above, LPR events were similar in the normal group and OSA patient group in other studies; therefore, further study on this is necessary. Second, the humidification function of CPAP also might have positive effects on LPR symptoms. Humidification can relieve dryness of the laryngeal mucosa, and promote overall tissue healing and mucosal homeostasis [17]. CPAP also prevents dryness of the laryngeal mucosa by reducing mouth leakage during sleep. Finally, animal models have shown that repeated pressure changes in the upper airway and repeated collapse and reopening trigger the inflammatory process in the upper airway [18]. This inflammatory process is thought to cause tissue changes, leading to the upper respiratory tract damage observed in OSA patients, and positive airway pressure is expected to improve this damage.

There are several limitations to our study. First, we assessed LPR using both a subjective patient-reported questionnaire and endoscopic findings, rather than more objective measurements such as 24 h double-probe pH monitoring. To evaluate LPR as objectively as possible, three otolaryngologists not involved in the rest of the study evaluated RFS and showed moderate reliability (κ = 0.43). Second, this study was conducted as a retrospective study, and because of this, people who did not meet the inclusion criteria were excluded to have a small sample size, and a small number of females were included. Additionally, this study was not conducted in a blinded fashion and did not include a control group. In future studies, OSA patients with LPR being treated with PPIs should be enrolled and compared with a CPAP treatment group.

## 5. Conclusions

CPAP for OSA treatment significantly reduced laryngeal reflux symptoms and endoscopic findings in the larynx. To clarify these results, a larger-scale study with more patients and longer follow-up is needed. Additional studies using objective measures to evaluate the effects of CPAP on LPR are also warranted.

## Figures and Tables

**Table 1 jcm-10-02861-t001:** Reflux symptom index questionnaire.

Within the Last Month, How Did the Following Problem Affect You?	0 = No Problem 5 = Severe Problem					
Hoarseness or a problem with your voice	0	1	2	3	4	5
Clearing your throat	0	1	2	3	4	5
Excess throat mucus or postnasal drip	0	1	2	3	4	5
Difficulty swallowing food, liquids or pills	0	1	2	3	4	5
Coughing after you ate or after lying down	0	1	2	3	4	5
Breathing difficulties or choking episodes	0	1	2	3	4	5
Troublesome or annoying cough	0	1	2	3	4	5
Sensation of something sticking in your throat or a lump in your throat	0	1	2	3	4	5
Heartburn, chest pain, indigestion or stomach Acid coming up	0	1	2	3	4	5
Total	0	1	2	3	4	5

**Table 2 jcm-10-02861-t002:** Reflux finding score.

Subglottic edema	0 = absent 2 = present
Ventricular obliteration	2 = partial 4 = complete
Erythema/hyperemia	2 = arytenoid only 4 = diffuse
Vocal fold edema	1 = mild 2 = moderate 3 = severe 4 = polypoid
Diffuse laryngeal edema	1 = mild 2 = moderate 3 = severe 4 = obstructing
Posterior commissure hypertrophy	1 = mild 2 = moderate 3 = severe 4 = obstructing
Granuloma, granulation	0 = absent 2= present
Thick laryngeal mucus	0 = absent 2= present
Total	

**Table 3 jcm-10-02861-t003:** Characteristics of obstructive sleep apnea patients (*n* = 46).

Characteristics	Value
Age (yr)	52.70 ± 11.62
Sex (M:F)	43:3
BMI (kg/m^2^)	28.60 ± 7.16
AHI	49.53 ± 22.69
AHI after CPAP use	2.27 ± 1.80
CPAP compliance (h)	5.61 ± 1.39
CPAP compliance > 4 h	42 (91.3%)
Pre CPAP RSI (>13)	12 (26.1%)
Pre CPAP RFS (>7)	26 (56.5%)

**Table 4 jcm-10-02861-t004:** Comparison of individual reflux symptom index subdomains before and after CPAP treatment.

Variable	Reflux Symptom Index		*p* Value
	Pre CPAP	Post CPAP	
Hoarseness	1.26 ± 1.3	1.00 ± 1.03	0.223
Throat clearing	1.97 ± 1.40	1.19 ± 1.30	<0.001
Throat mucous	2.52 ± 1.50	1.55 ± 1.36	<0.001
Dysphagia	0.26 ± 0.51	0.23 ± 0.50	0.662
Coughing	0.42 ± 0.92	0.48 ± 0.81	0.690
Breathing difficulty	1.32 ± 1.40	0.81 ± 0.95	0.027
Troublesome cough	0.74 ± 1.15	0.42 ± 0.99	0.023
Foreign body sensation	1.29 ± 1.27	0.74 ± 1.23	<0.001
Heartburn sensation	1.03 ± 1.22	0.74 ± 0.96	0.095

**Table 5 jcm-10-02861-t005:** Comparison of individual reflux finding score subdomains before and after CPAP treatment.

Variable	Reflux Finding Score		*p* Value
	Pre CPAP	Post CPAP	
Subglottic edema	0.30 ± 0.70	0.13 ± 0.50	0.031
Ventricular obliteration	0.91 ± 1.15	0.50 ± 0.84	0.013
Erythema	2.57 ± 0.98	1.65 ± 0.85	<0.001
Vocal fold edema	0.98 ± 0.80	0.59 ± 0.50	<0.001
Diffuse laryngeal edema	0.89 ± 0.67	0.57 ± 0.50	0.001
Posterior commissure hypertrophy	1.15 ± 0.81	0.89 ± 0.77	0.060
Granuloma	0.10 ± 0.43	0.07 ± 0.33	0.420
Thick endolaryngeal mucus	0.72 ± 0.96	0.26 ± 0.68	0.004

**Table 6 jcm-10-02861-t006:** Changes in reflux symptom index scores and reflux finding scores.

Variable	RSI Score Difference	*p* Value	RFS Difference	*p* Value
Severity of OSA		0.37		0.99
Moderate (*n* = 12)	4.25 ± 4.94		3.33 ± 2.57	
Severe (*n* = 34)	3.50 ± 3.54		2.70 ± 2.48	
Body mass index		0.63		0.30
Normal (*n* = 11)	3.00 ± 2.87		2.54 ± 2.62	
Overweight (*n* = 25)	4.50 ± 4.53		2.76 ± 2.55	
Obese (*n* = 10)	2.71 ± 2.92		4.00 ± 2.58	

## Data Availability

Data sharing not applicable.

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
