# Peer review of "Potential Therapeutic Effect of Continuous Positive Airway Pressure on Laryngopharyngeal Reflux in Obstructive Sleep Apnea Patients"

_jcm, 2021, doi:10.3390/jcm10132861_

Round 1

Reviewer 1 Report

The important limits are clearly presented and should be the base for future projects on the subject. An objective diagnosis of LPR/GERD via gastroenterological thorough evaluation and exams is needed, also to better understand the underlying physiopathology of OSA-related LPR. 

Author Response

Thank you for taking the time to write us a review. We also hope that the pathophysiology of OSA-related LPR will be elucidated through further studies. 

Reviewer 2 Report

This small retrospective study evaluated the change in laryngeal reflux symptoms and physical findings following treatment of obstructive sleep apnea with CPAP.  The data is presented clearly and the results support the authors' conclusions.

There are some minor points for the authors to address:

  1. The sentence in lines 127- 128 is missing a word or words.
  2. it is not clear what the sentence in lines 165-166 means. This needs to be clarified.
  3. In the limitations paragraph, the authors do not address the small sample size and the retrospective nature of the study design. They also did not discuss the small number of females in the analysis and the potential for bias given this. 
  4. The last sentence in the conclusions section is unclear- ... objective measures to evaluate the effects of CPAP (on what?) are also warranted.

Author Response

Thank you for taking the time to write us a review

Point 1. : We revised the sentence to indicate that some individual variables differ significantly

Point 2. : We revised the sentence to indicated that reduced mouth leakage prevent dryness of laryngeal mucosa

Point 3. : We add our limitation of the small sample size and the small number of females in the analysis

Point 4. : We revised the sentence to indicated that objective measures to evaluate the effects of CPAP on LPR are also warranted